# Detection of bisphenol A in thermal paper receipts and assessment of human exposure: A case study from Sharjah, United Arab Emirates

**Lucy Semerjian** *, **Najla Alawadhi**, **Khulud Nazer**

Department of Environmental Health Sciences, College of Health Sciences, University of Sharjah, Sharjah, United Arab Emirates

* lsemerjian@sharjah.ac.ae

**Data Availability Statement:** All relevant data are within the paper.

**Funding:** This research was thankfully funded by Office of Dean, College of Health Science at

## Abstract

Bisphenol A (BPA) is an industrial chemical that is widely used in various industrial applications. It has concerns in its use as a color developer in thermal paper receipts since it is identified as an endocrine disruptor and causes hormonal disturbances. In this study, thirty thermal paper receipt samples were randomly collected from various locations in Sharjah, United Arab Emirates and analyzed using high performance liquid chromatography–mass spectrometry. Sixty percent (60%) of receipt samples showed BPA levels above the acceptable limit (200 ng/mg) set by the European Union (EU) for thermal papers. On the other hand, 40% of the samples reported very low BPA levels (< 0.02 ng/mg). However, estimated weight adjusted daily intakes (EDI) ranged between $8.22 \times 10^{-11}$ and 0.000812 µg/kg bw/day for the general population, and between $7.89 \times 10^{-9}$ and 0.0681 µg/kg bw/day for the occupationally exposed cashiers. Thus, all calculated EDIs were below the European Food Safety Authority Tolerable Daily Intake (4 µg/kg·bw/day) and the provisional Health Canada Tolerable Daily Intake (25 ug/kg bw/day) under varying paper-to-skin transfer coefficients and absorption fractions. Nevertheless, due to its health effects and recent legal restrictions by EU, the occurrence of co-exposure to dietary and non-dietary sources should be considered in the health risk assessment of Bisphenol A, mainly for people with frequent occupational exposure to thermal paper, and especially with the increased use of sanitizers. The current study is a first within the UAE context in relation to BPA in thermal paper receipts, thus its significance especially with the recent EU enforcement of BPA limits in paper receipts. The study highlights that proper policies as well as education and awareness may assist in limiting transdermal BPA exposure for the general and occupationally exposed populations.

## Introduction

Bisphenol A (BPA) ($C_{15}H_{16}O_2$; 2,2-bis(4-hydroxyphenyl) propane) is a synthetic xenoestrogen. It is widely used in varying industrial fields such as polycarbonate plastic production of

University of Sharjah. The funders had no role in study design, data collection and analysis, decision to publish, or preparation of the manuscript.

**Competing interests:** The authors have declared that no competing interests exist.

food and beverage containers, epoxy resin linings, paints and coatings, glazing in building and construction and aerospace. BPA can also be present in thermal papers as an additive in its unreacted form to serve as a color developer [1, 2]. BPA could get into the human body through diet, inhalation and dermal contact. Due to similar structure to normal hormones, BPA can imitate the natural hormones and cause hormonal system disturbance. BPA can occur in two chemical forms: the conjugated form enters the body by oral route and the unconjugated form via dermal contact and inhalation. Only the unconjugated form can bind to estrogen receptors thus making it more active and hazardous [3]. Prolonged exposure to high concentrations of BPA can cause health effects, mainly causing damage to the reproductive and hormonal systems or causing cardiovascular diseases, development abnormalities and insulin resistance [4, 5]. BPA can also cause contamination of the environment, including surface waters and sediments. Among polycarbonate bottles, canned food and beverages, PVC food packaging stretch films, and dental sealants, thermal papers are still one of the main sources of BPA [6]. Moreover, BPA in thermal paper is in its monomeric form, which can result in its easy transfer to the human body and penetrates deep in the skin making it very hard to be washed off. Furthermore, a wet or greasy skin allows up to ten-fold higher transfer of BPA [6, 7].

In recent years, BPA in thermal receipt papers has gained significant public attention as an environmental pollutant by both the European Union (EU) and the US National Toxicology Program (USNTP) [8]. From an occupational perspective, cashiers have a higher dermal exposure to BPA as they handle receipts frequently throughout their working days. Recent studies have shown that the mean amount of total BPA in urine was 2.5 times higher for people with regular contact with thermal paper than for control groups [6]. Thus, to prevent exposure to BPA from receipts, the European Union set a limit of <0.02% by weight (0.2 mg/g, 200 μg/g or ng/mg) for BPA concentration in thermal papers and the restriction has been in force since January 2020 [9].

Findings from various studies conducted worldwide revealed varying concentrations of BPA in thermal receipt papers, some in exceedance of EU BPA limit. Conducted risk assessments also revealed a variation in estimated daily intakes (EDI) of BPA through dermal contact with thermal receipts. Frankowski et al. [6] investigated the presence of BPA in 220 samples of thermal papers collected from several countries. Results showed that BPA concentrations in 22% of investigated samples exceeded the EU BPA limit of 0.2 mg/g. Another study conducted by Molina-Molina et al. [10] to explore BPA and Bisphenol S (BPS) concentrations in thermal papers (n = 112) from Brazil, France, and Spain revealed that BPA was found in 95.3% of receipts from Spain, 90.9% from Brazil, and 51.1% from France at concentrations up to 20.3 mg/g of paper. Wong et al. [11] investigated the presence of BPA in 30 thermal paper receipt samples collected from various stores in British Columbia, Canada; thirteen (13) samples contained BPA levels between 0.124–871.17 mg/kg paper. In China, Zhang et al. [12] collected thermal paper samples from different sources; all samples exhibited BPA contents exceeding the allowable limit as the BPA content was 0.696 mg/g in a supermarket receipt, 0.937 mg/g in a laundry receipt, and ranged between 0.306–3.780 mg/g in three taxi tickets. In Spain, thermal printing paper samples were collected from different establishments by Castro et al. [13]; results showed that BPA concentrations varied from 0.005 to 6.30%. Moreover, in a study conducted in Turkey by Yalcin et al. [14], BPA was found in twelve thermal paper receipts at a range between 0.40±0.11 mg/g and 21.65 ± 0.83 mg/g. In a study conducted in Belgium by Geens et al. [15], BPA was detected at concentrations above 0.9% in 73% of investigated thermal paper receipts (n = 44) which resulted in an estimated daily intake (EDI) of 6.4 ng/kg bw/day for the general population based on a dermal absorption rate of 27%, reported by Biedermann et al. [16]. Concentrations of BPA in supermarkets receipts were also

investigated by Lu et al. [17] in Shenzhen, China; BPA was detected in all analyzed samples (n = 42) with concentrations ranging from 2.58 to 14.7 mg/g. Russo et al. [18] monitored BPA and BPS in thermal paper receipts (n = 50) from Italian markets; BPA was found in 44 samples at a concentration of 107.47 μg/100 mg of paper. In the same study, a risk assessment was conducted for the transdermal route; daily intake was estimated to be 66.8 μg/day for BPA for occupationally exposed people, namely cashiers. Moreover, in a study conducted by [15], concentrations of BPA were determined in 15 types of paper products collected from several countries. BPA was found in 94% of thermal receipt papers at concentrations ranging from below the limit of quantitation (LOQ, 1 ng/g) to 13.9 mg/g. The majority (81%) of other paper products contained BPA at concentrations ranging from below the LOQ to 14.4 μg/g which made it evident that thermal receipt papers tend to contain higher concentrations of BPA. The same study calculated the daily intake of BPA through dermal absorption to be 17.5 ng/day for the general population and 1,300 ng/day for occupationally exposed people.

Although in many studies, the recorded BPA levels as well as the calculated EDIs are minor compared with exposure to BPA through diet (oral pathway); nevertheless, dermal exposure needs to be considered with the overall risk assessment of BPA exposure and specifically through thermal paper handling as thermal papers contribute to the majority of dermal BPA exposure. A study conducted by [6] recorded elevated geometric mean (GM) urinary BPA levels for cashiers after two continuous hours of work without gloves. The geometric mean urinary BPA concentration was 1.8 mg/L before exposure and 5.8 mg/L after exposure. Also, in another study [19], urinary BPA levels were significantly higher in female cashiers (GM 5.45 mg/L) compared to females with unlikely occupational exposure (GM 2.16 mg/L). Moreover, several studies revealed that sanitizers and skincare products can increase the dermal penetration of lipophilic compounds, such as BPA, up to 100-fold [6, 16, 20]. Thus, in view of the current COVID-19 pandemic where sanitizer use rates are at their highest, and especially by receipt/currency handling cashiers, the dermal exposure to BPA in occupational settings is expected to further increase.

The driver of the current research study results from lack of published data on BPA concentrations in thermal paper receipts used in United Arab Emirates albeit the availability of numerous global studies. Therefore, this research aims to investigate BPA concentrations in thermal paper receipts circulating in Sharjah, United Arab Emirates and calculate estimated daily intakes of recorded BPA levels in occupational (cashiers) settings and for general population. Findings will assist in raising awareness and sharing knowledge related to BPA exposure through thermal paper receipts, and in protecting public and environmental health.

## Materials and methods

### Sample collection

A total of 30 thermal paper receipt samples were randomly collected from various locations in Sharjah, UAE including supermarkets (n = 11), restaurants (n = 6), automatic teller machines (ATMs) (n = 6) and others (n = 7) such as receipts from a hospital, a clothing store, a laundry, delivery service, a tailor shop, and a pharmacy.

### Sample extractions and instrumental analysis

Individual thermal papers were cut into small pieces and 25mg of each paper sample was extracted into 50ml of pure distilled water for 60 minutes at room temperature according to a procedure reported by [14]. The supernatants were separated and the extracts were analyzed using high performance liquid chromatography/ mass spectrometry (HPLC-MS) technique.

The high performance liquid chromatography- mass spectrometry analysis was carried out using an Acquity UPLC (Waters, USA) connected to ESI Xevo TQD (Waters, USA). Sample volumes of 10 uL were injected into a Zorbax Eclipse plus C18 column (Agilent, 50 mm 2.1 mm I.D.; 2.1 mm) maintained at 40°C. The mobile phase employed in the analysis consisted of A:water and B:Methanol at a flow rate of 0.3 mL/min in the following gradient: 0 min 35% B; 3 min 95% B; 4.2 min 95% B and 4.5 min 35% B. The chromatographic column effluent was directed to the mass spectrometer through the electrospray ionization source which operated in a negative ion mode MRM of 227.1 to 211 (quantitative) and 227.1 to 133 (qualitative). The dwell time for each mass transition detected in the multiple reaction monitoring mode was set to 25 ms. All compounds were detected using the following settings for the ion source and mass spectrometer: desolvation gas flow 650 L/hr, source temperature 350°C, capillary voltage -1000V, and cone voltage-31V. In this study, the external standard calibration method was used. Standard stock solutions were prepared in methanol and were further diluted to obtain working standard solutions at concentrations of 200, 350, 500, 750, and 1000 ng/ml.

## Chemicals and reagents

The analytical standard of BPA (> 99% purity) was obtained from Sigma-Aldrich (Saint Louis, USA), and methanol (MeOH) solvent, chromatographic grade, was purchased from J. T. Baker (Phillipsburg, USA). High purity deionized water (Millipore RiOs-DITM, Bedford, USA) was used for all the experiments.

## BPA exposure assessment

There are different exposure routes that can bring BPA into the human body such as ingestion, inhalation of contaminated air and dermal absorption. However, dermal exposure remains the highest plausible exposure pathway in relation to BPA exposure through thermal paper handling. Thus, in this study the exposure of Sharjah, UAE population to BPA via handling of thermal receipt papers was evaluated according to the method reported by [21], and the following equation (Eq 1) was used to calculate weight adjusted estimated daily intakes through dermal exposure for both the general population and occupationally exposed cashiers. The concentrations of BPA in thermal papers, the time that they are in contact with skin, the surface area of skin in contact with the papers, the rate of transfer from the paper to the skin surface, the number of handling events experienced daily are all variables that need to be considered to assess human BPA intake via thermal paper handling. In the absence of UAE-specific data on selected variables, literature-based data was used and EDI values were calculated for both the general population and the occupationally exposed cashiers under all plausible scenarios.

$$EDI\ (\mu g/day/kg) = [k \times C \times HF \times HT \times AF /10^9]/BW \tag{1}$$

Where $k$ (ng/s) is the paper-to-skin transfer coefficient of BPA estimated as 1072, 1838, or 21522 [3]. Reported $k$ values varied as conducted extractability experiments exhibited varying coefficients based on skin conditions being dry, wet, or very greasy [3]

C (μg/g) is the concentration of BPA in the thermal paper samples;

HF (times/day) is the handling frequency;

HT (s/time) is the handling time of paper; and

AF is the absorption fraction of BPA by skin, which is 2.3, 8.6 or 27% [3]. A wide range of absorption factors have been reported in previous studies as they are calculated from different experimental systems: 2.3–8.6% from skin explants, and 27% from live hands after application of BPA [3].

**Table 1. BPA concentrations in thermal papers investigated in the current study and in thermal papers from similar sources reported in global studies.**

| Sources of receipts | BPA (ng/mg) in the current study | BPA (ng/mg) in other studies | References |
|---|---|---|---|
| Supermarkets (n = 11) | <0.02–854.2 | 2,580–14,700 | [17] |
| | | 696 | [12] |
| | | 7,590–21,650 | [14] |
| Restaurants (n = 6) | <0.02–978.2 | 16,900–20,000 | [15] |
| ATMs (n = 6) | <0.02–776.2 | 15,000 | [14] |
| | | 1,000–17,900 | [15] |
| *Others (n = 7)* | <0.02–931.4 | 937 (laundry receipts) | [12] |

BW is body weight (Kg), assigned by European Commission at default body weight of 60 Kg for adult females, and 70 Kg for adult males [22].

For the general population, the handling time and handling frequency of thermal paper receipts were estimated at 5 seconds per time, and 2 times per day, respectively. As for individuals working as cashiers (in supermarkets only) who handle receipts frequently the following estimates were used in this study for a work shift of 8 hours per day: HT was estimated at 10 seconds per time, and HF was estimated at an average of 96 times per day (i.e., a customer every 5 min). Similar estimates were adopted in previous comparable research [17, 23].

## Results and discussion

### Concentrations of BPA

BPA was detected in sampled thermal receipt papers at concentrations ranging between <0.02 ng/mg to 978.2 ng/mg as reported in Table 1. Concentrations of BPA detected in investigated sources of receipts were lower in this study in comparison with other global studies which also investigated BPA levels in thermal paper receipts from similar sources [12, 14, 15, 17]. The highest maximum recorded concentrations were for restaurants receipts, and lowest maximum concentrations were for ATM receipts.

Findings from global studies conducted to investigate BPA levels in thermal paper receipts from varying sources, thus not necessarily from supermarket, restaurant, or ATM receipts exhibited varying concentrations of BPA as well as summarized in Table 2. It is evident that maximum concentrations of BPA detected in investigated sources of receipts remain lower in this study in comparison with other global studies except for levels reported by [11].

BPA concentrations in all investigated samples are shown in Fig 1 in comparison to the maximum allowable limit for BPA (200 ng/mg) as set by the [9] and in force since 2020. Fig 1 exhibits that 54.6% (n = 6) of thermal paper samples collected from supermarkets, 50% (n = 3) of thermal paper samples collected from restaurants, 83.3% (n = 5) of thermal paper samples

**Table 2. BPA concentrations in thermal paper receipts from varying sources in current study and various global studies.**

| BPA concentration (ng/mg) | References |
|---|---|
| <0.02–978.2 | Current study |
| 8,300–17,100 | [24] |
| 0.124–871.17 | [11] |
| 50 to 63,000 | [13] |
| <LOQ—15,337.3 (mean = 1,074.7) | [10] |
| < 0.01–13,900 | [15] |

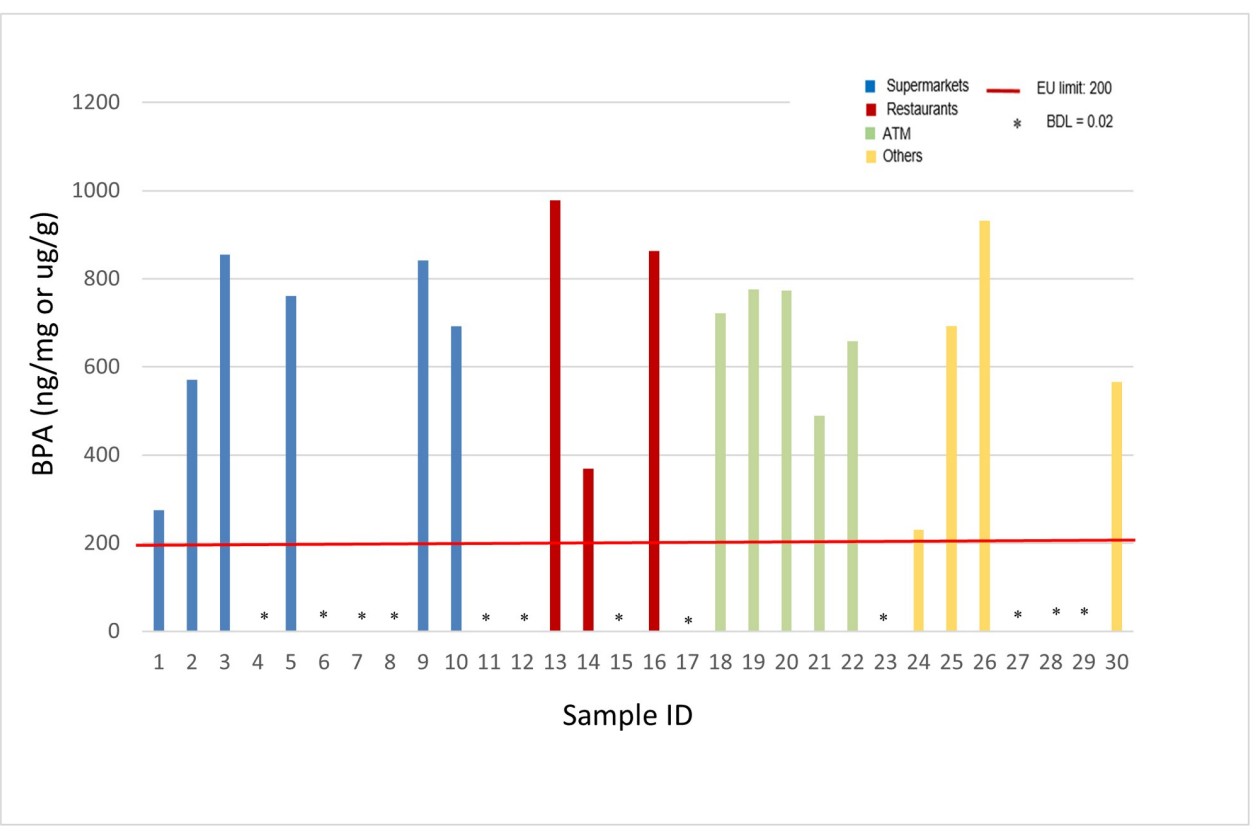

**Fig 1. BPA concentrations in investigated thermal receipts.**

collected from ATMs, and 57.1% (n = 4) of thermal paper samples collected from other locations exceed the EU limit. On the other hand, 40% of all samples exhibited BPA levels below detection limit of 0.02 ng/mg and thus below the BPA EU limit.

## Weight adjusted estimated daily intakes

Weight adjusted estimated daily intakes from dermal BPA exposure through thermal papers for each source of thermal paper were calculated for both males and females within the general population as well as for cashiers who are at higher risk of exposure through dealing with receipts more frequently and for longer periods. All plausible scenarios for contributing factors, namely $k$ (paper-to-skin transfer coefficient) and AF (absorption factor) compiled from similar global research were considered in the current study as detailed in Materials and Methods section. Tables 3–6 summarize ranges of EDIs for the general population (females and males) from dermal BPA exposure through thermal papers for each source of thermal paper in the respective order of supermarkets, restaurants, ATMs, and others. In addition, ranges of weight adjusted EDIs for cashiers (females and males) from occupational dermal BPA exposure through thermal papers from supermarkets only are summarized in Table 7.

It is evident from calculated EDIs that under varying $k$ and AF values, females exhibit higher EDIs compared to males for both the general population as well as cashiers. For the general population, EDIs are higher from exposure to thermal receipts originating from restaurants, followed by other receipts, supermarket receipts, and ATM receipts, respectively. It is expected to observe as $k$ and AF values increase, calculated EDIs increase. In comparison to

**Table 3. EDI (µg/kg bw/day) range for general population (females and males) from "supermarket" thermal paper receipts under varying AF and *k* values.**

| | EDI (Females) | | |
|---|---|---|---|
| | **K = 1,072** | **K = 1,838** | **K = 21,522** |
| **AF = 2.3%** | $8.22 \times 10^{-11}$–$3.51 \times 10^{-6}$ | $1.41 \times 10^{-10}$–$6.02 \times 10^{-6}$ | $1.65 \times 10^{-9}$–$7.05 \times 10^{-5}$ |
| **AF = 8.6%** | $3.07 \times 10^{-10}$–$1.31 \times 10^{-5}$ | $5.27 \times 10^{-10}$–$2.25 \times 10^{-5}$ | $6.17 \times 10^{-9}$–$0.000263505$ |
| **AF = 27%** | $9.65 \times 10^{-10}$–$4.12 \times 10^{-5}$ | $1.65 \times 10^{-9}$–$7.07 \times 10^{-5}$ | $1.94 \times 10^{-8}$–$0.000827284$ |
| | **EDI (Males)** | | |
| **AF = 2.3%** | $7.04 \times 10^{-11}$–$3.00 \times 10^{-6}$ | $1.21 \times 10^{-10}$–$5.16 \times 10^{-6}$ | $1.41 \times 10^{-9}$–$6.04 \times 10^{-5}$ |
| **AF = 8.6%** | $2.63 \times 10^{-10}$–$1.13 \times 10^{-5}$ | $4.52 \times 10^{-10}$–$1.93 \times 10^{-5}$ | $5.29 \times 10^{-9}$–$0.000225862$ |
| **AF = 27%** | $8.27 \times 10^{-10}$–$3.53 \times 10^{-5}$ | $1.42 \times 10^{-9}$–$6.05 \times 10^{-5}$ | $1.66 \times 10^{-8}$–$0.000709101$ |

HF = 2; HT = 5 sec/time; BW = 60 or 70 Kg; $C_{min}$ = 0.02 µg/g and $C_{max}$ = 854.2 µg/g

**Table 4. EDI (µg/kg bw/day) range for general population (females and males) from "restaurant" thermal paper receipts under varying AF and *k* values.**

| | EDI (Females) | | |
|---|---|---|---|
| | **K = 1,072** | **K = 1,838** | **K = 21,522** |
| **AF = 2.3%** | $8.22 \times 10^{-11}$–$4.02 \times 10^{-6}$ | $1.41 \times 10^{-10}$–$6.89 \times 10^{-6}$ | $1.65 \times 10^{-9}$–$8.07 \times 10^{-5}$ |
| **AF = 8.6%** | $3.07 \times 10^{-10}$–$1.50 \times 10^{-5}$ | $5.27 \times 10^{-10}$–$2.58 \times 10^{-5}$ | $6.17 \times 10^{-9}$–$0.000301757$ |
| **AF = 27%** | $9.65 \times 10^{-10}$–$4.72 \times 10^{-5}$ | $1.65 \times 10^{-9}$–$8.09 \times 10^{-5}$ | $1.94 \times 10^{-8}$–$0.000947377$ |
| | **EDI (Males)** | | |
| **AF = 2.3%** | $7.04 \times 10^{-11}$–$3.45 \times 10^{-6}$ | $1.21 \times 10^{-10}$–$5.90 \times 10^{-6}$ | $1.41 \times 10^{-9}$–$6.91 \times 10^{-5}$ |
| **AF = 8.6%** | $2.63 \times 10^{-10}$–$1.29 \times 10^{-5}$ | $4.52 \times 10^{-10}$–$2.21 \times 10^{-5}$ | $5.29 \times 10^{-9}$–$0.000258649$ |
| **AF = 27%** | $8.27 \times 10^{-10}$–$4.04 \times 10^{-5}$ | $1.42 \times 10^{-9}$–$6.93 \times 10^{-5}$ | $1.66 \times 10^{-8}$–$0.000812037$ |

HF = 2; HT = 5 sec/time; BW = 60 or 70 Kg; $C_{min}$ = 0.02 µg/g and $C_{max}$ = 978.2 µg/g

**Table 5. EDI (µg/kg bw/day) range for general population (females and males) from "ATMs" thermal paper receipts under varying AF and *k* values.**

| | EDI (Females) | | |
|---|---|---|---|
| | **K = 1,072** | **K = 1,838** | **K = 21,522** |
| **AF = 2.3%** | $8.22 \times 10^{-11}$–$3.18 \times 10^{-6}$ | $1.40 \times 10^{-10}$–$5.46 \times 10^{-6}$ | $1.65 \times 10^{-9}$–$6.40 \times 10^{-5}$ |
| **AF = 8.6%** | $3.07 \times 10^{-10}$–$1.19 \times 10^{-5}$ | $5.268 \times 10^{-10}$–$2.04 \times 10^{-5}$ | $6.16 \times 10^{-9}$–$0.000239444$ |
| **AF = 27%** | $9.64 \times 10^{-10}$–$3.74 \times 10^{-5}$ | $1.65 \times 10^{-9}$–$6.42 \times 10^{-5}$ | $1.93 \times 10^{-8}$–$0.000751742$ |
| | **EDI (Males)** | | |
| **AF = 2.3%** | $7.04 \times 10^{-11}$–$2.73 \times 10^{-6}$ | $1.21 \times 10^{-10}$–$4.69 \times 10^{-6}$ | $1.41 \times 10^{-9}$–$5.49 \times 10^{-5}$ |
| **AF = 8.6%** | $2.63 \times 10^{-10}$–$1.02 \times 10^{-5}$ | $4.52 \times 10^{-10}$–$1.75 \times 10^{-5}$ | $5.29 \times 10^{-9}$–$0.000205237$ |
| **AF = 27%** | $8.27 \times 10^{-10}$–$3.21 \times 10^{-5}$ | $1.42 \times 10^{-9}$–$5.50 \times 10^{-5}$ | $1.66 \times 10^{-8}$–$0.00064435$ |

HF = 2; HT = 5 sec/time; BW = 60 or 70 Kg; $C_{min}$ = 0.02 µg/g and $C_{max}$ = 776.2 µg/g

the general population, cashiers recorded higher EDIs, both for females and males, under varying *k* and AF values given the higher contact frequency and time. Nevertheless, all calculated EDIs remain below the Tolerable Daily Intake (TDI) of 4 µg/kg·bw/day established by the European Food Safety Authority (EFSA) [25], and the provisional TDI of 25 ug/kg bw/day by Health Canada [11]. Solely the EDIs for cashiers calculated at a *k* value of 21,522 and AF value of 27% (aggressive scenario) exceeded the proposed amendment to the maximum allowable dose level (MADL) of BPA (0.052 µg/kg·bw/day for females and 0.043 µg/kg·bw/day for

**Table 6. EDI (μg/kg bw/day) range for general population (females and males) from "others" thermal paper receipts under varying AF and $k$ values.**

| | EDI (Females) | | |
|---|---|---|---|
| | **K = 1,072** | **K = 1,838** | **K = 21,522** |
| **AF = 2.3%** | $8.22\times 10^{-11}$-$3.83\times 10^{-6}$ | $1.41\times 10^{-10}$-$6.56\times 10^{-6}$ | $1.65\times 10^{-9}$-$7.68\times 10^{-5}$ |
| **AF = 8.6%** | $3.07\times 10^{-10}$-$1.43\times 10^{-5}$ | $5.27\times 10^{-10}$-$2.45\times 10^{-5}$ | $6.16\times 10^{-9}$-$0.00028732$ |
| **AF = 27%** | $9.65\times 10^{-10}$-$4.49\times 10^{-5}$ | $1.65\times 10^{-9}$-$7.70\times 10^{-5}$ | $1.93\times 10^{-8}$-$0.000902052$ |
| | EDI (Males) | | |
| **AF = 2.3%** | $7.04\times 10^{-11}$-$3.28\times 10^{-6}$ | $1.21\times 10^{-10}$-$5.62\times 10^{-6}$ | $1.41\times 10^{-9}$-$6.59\times 10^{-5}$ |
| **AF = 8.6%** | $2.63\times 10^{-10}$-$1.23\times 10^{-5}$ | $4.57\times 10^{-10}$-$2.10\times 10^{-5}$ | $5.29\times 10^{-9}$-$0.000246274$ |
| **AF = 27%** | $8.27\times 10^{-10}$-$3.85\times 10^{-5}$ | $1.42\times 10^{-9}$-$6.60\times 10^{-5}$ | $1.66\times 10^{-8}$-$0.000773187$ |

HF = 2; HT = 5 sec/time; BW = 60 or 70 Kg; $C_{min}$ = 0.02 μg/g and $C_{max}$ = 931.4μg/g

males) through dermal exposure to solid materials by California's Office of Environmental Health Hazard Assessment [26]. Moreover, calculated EDIs for the general population in the current study under varying $k$ and AF values and for the various investigated sources of thermal receipts remain lower in comparison to calculated EDIs for general population in other studies recorded at 0.000892 ug/kg bw/day [18] and 0.0064 ug/kg bw/day [15]. Similarly, calculated EDIs for the cashiers in the current study remain lower in comparison to calculated EDIs for the occupationally exposed in other studies recorded at 0.954 ug/kg bw/day [18] and 0.0217 ug/kg bw/day [15] except when considering a $k$ value of 21,522, and AF value of 27%.

Therefore, no health risks are imposed by BPA levels recorded in the current study for the general population as well as occupationally exposed cashiers. Although in many studies, including the current study, recorded BPA levels as well as the calculated EDIs are minor compared with exposure to BPA through diet (oral pathway); nevertheless, dermal exposure needs to be considered with the overall risk assessment of BPA exposure and specifically through thermal paper handling as thermal papers contribute to the majority of dermal BPA exposure, especially with the increased use of sanitizers in view of COVID-19 as several studies revealed that sanitizers and skincare products can increase the dermal penetration of lipophilic compounds, such as BPA, up to 100 fold [6, 16, 20].

## Conclusions

The current study sheds new light for the first time on the estimated transdermal BPA intakes from thermal paper receipts in the population of Sharjah, United Arab Emirates. Sixty percent

**Table 7. EDI (μg/kg bw/day) range for cashier (females and males) from "supermarket" thermal paper receipts under varying AF and k values.**

| | EDI (Females) | | |
|---|---|---|---|
| | **K = 1,072** | **K = 1,838** | **K = 21,522** |
| **AF = 2.3%** | $7.89\times 10^{-9}$-$0.000336978$ | $1.41\times 10^{-8}$-$0.000577767$ | $1.58\times 10^{-7}$-$0.006942129$ |
| **AF = 8.6%** | $2.95\times 10^{-8}$-$0.001260007$ | $5.06\times 10^{-8}$-$0.002160347$ | $5.92\times 10^{-7}$-$0.025296511$ |
| **AF = 27%** | $9.26\times 10^{-8}$-$0.003955834$ | $1.59\times 10^{-7}$-$0.006782485$ | $1.86\times 10^{-6}$-$0.079437874$ |
| | EDI (Males) | | |
| **AF = 2.3%** | $6.76\times 10^{-9}$-$0.000288839$ | $1.21\times 10^{-8}$-$0.000495229$ | $1.36\times 10^{-7}$-$0.005950396$ |
| **AF = 8.6%** | $2.53\times 10^{-8}$-$0.001080006$ | $4.34\times 10^{-8}$-$0.001851726$ | $5.08\times 10^{-7}$-$0.021682724$ |
| **AF = 27%** | $7.94\times 10^{-8}$-$0.003390715$ | $1.36\times 10^{-7}$-$0.005813558$ | $1.59\times 10^{-6}$-$0.068089606$ |

HF = 96; HT = 10 sec/time; BW = 60 or 70 Kg; $C_{min}$ = 0.02 μg/g and $C_{max}$ = 854.2 μg/g

(60%) of receipt samples showed BPA levels above the acceptable limit (200 ng/mg) set by the European Union for thermal papers. On the other hand, 40% of the samples reported very low BPA levels ($<$ 0.02 ng/mg). Estimated weight adjusted daily intakes (EDI) ranged between $8.22 \times 10^{-11}$ and 0.000812 μg /kg bw/day for the general population, with the maximum EDI attributed to the paper receipts generated from restaurants. For the occupationally exposed cashiers, estimated weight adjusted daily intakes ranged between $7.89 \times 10^{-9}$ and 0.0681 μg/kg bw/day. All calculated EDIs remain below the EFSA TDI (4 μg/kg·bw/day) and the provisional Health Canada TDI (25 ug/kg bw/day).

## Recommendations

BPA remains the most common color developer used in thermal papers circulated in global markets. Proper policies encouraging the use of BPA-free paper receipts as well as education and awareness for cashiers and general population may assist in limiting transdermal BPA exposure. Information could be spread on how to avoid touching the receipts by wearing gloves especially with the excessive use of lotions and sanitizers, washing hands after handling of receipts, declining receipts, and avoiding children's handling of receipts. The best option will be shifting to electronic receipt options to protect both public health and environmental resources.

## Acknowledgments

The authors gratefully acknowledge the inputs from Amna AlKaabi and Haya Alami throughout the project. Special thanks to University of Sharjah Research Institute for Medical and Health Sciences, namely Dr. M. Harb Semreen, Badriah Ebrahim, Muath Khairi, and Yusur Almusleh for their assistance with the laboratory facilities, sample extraction and analysis procedures.

## Author Contributions

**Conceptualization:** Lucy Semerjian, Najla Alawadhi.

**Funding acquisition:** Lucy Semerjian.

**Investigation:** Najla Alawadhi, Khulud Nazer.

**Methodology:** Lucy Semerjian.

**Supervision:** Lucy Semerjian.

**Validation:** Lucy Semerjian.

**Writing – original draft:** Najla Alawadhi, Khulud Nazer.

**Writing – review & editing:** Lucy Semerjian.

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
