## [Decision Letter · Decision Letter 0]

23 Dec 2022

PONE-D-22-29094

Detection of Bisphenol A in Thermal Paper Receipts and Assessment of Human Exposure: A Case Study from Sharjah, United Arab Emirates

PLOS ONE

Dear Dr. Semerjian,

Thank you for submitting your manuscript to PLOS ONE. After careful consideration, we feel that it has merit but does not fully meet PLOS ONE’s publication criteria as it currently stands. Therefore, we invite you to submit a revised version of the manuscript that addresses the points raised during the review process.

We look forward to receiving your revised manuscript.

Kind regards,

Yasmina Abd‐Elhakim

Academic Editor

PLOS ONE

Journal Requirements:

  "The authors gratefully acknowledge the inputs from Amna AlKaabi and Haya Alami throughout the project. Special thanks to University of Sharjah Research Institute for Medical and Health Sciences, namely Dr. M. Harb Semreen, Badriah Ebrahim, Muath Khairi, and Yusur Almusleh for their assistance with the laboratory facilities, sample extraction and analysis procedures. This research was thankfully funded by Office of Dean, College of Health Science at University of Sharjah."

  "This research was thankfully funded by Office of Dean, College of Health Science at University of Sharjah. The funders had no role in study design, data collection and analysis, decision to publish, or preparation of the manuscript."

Reviewers' comments:

Reviewer's Responses to Questions

**Comments to the Author**

1. Is the manuscript technically sound, and do the data support the conclusions?

Reviewer #1: Partly

Reviewer #2: Yes

Reviewer #3: Yes

Reviewer #4: Yes

2. Has the statistical analysis been performed appropriately and rigorously? 

Reviewer #1: I Don't Know

Reviewer #2: Yes

Reviewer #3: Yes

Reviewer #4: Yes

3. Have the authors made all data underlying the findings in their manuscript fully available?

Reviewer #1: Yes

Reviewer #2: Yes

Reviewer #3: Yes

Reviewer #4: Yes

4. Is the manuscript presented in an intelligible fashion and written in standard English?

Reviewer #1: Yes

Reviewer #2: Yes

Reviewer #3: No

Reviewer #4: No

5. Review Comments to the Author

Reviewer #1: Dear author,

abstract is vague, please rewrite to be clear for readers.

introduction is too long

number of paper is insufficient.

tables unclear, specially table 3

discussion must be supported by extra references.

with best regards,

Reviewer #2: The conclusion should be more specific. It should only contain the important findings. The discussion section can include the comments made in the conclusion.

It would enrich the study if the study participants' health problems attributable to BPA toxicity were investigated.

There are a few grammatical/typographical mistakes.

Reviewer #3: The paper presents the results of studies Detection of Bisphenol A in

Thermal Paper Receipts and Assessment of Human Exposure: A Case Study

from Sharjah, United Arab Emirates. The issue raised in the paper is

important and the manuscript falls within the scope of plos one. The

paper would be of interest to the readers of the journal; however, I

recommend this paper undergo major revisions. The results are quite

interesting but still the flaws are needed to be addressed.

Introduction is too short and simple and written in very simple and

unscientific way

I think that at the end of the abstract should be inserted a brief

conclusion to help readers in getting a better understanding of what

is the significance of this study. What progress against the most

recent state-of-the-art similar studies was made?

Quantitative information should be provided in the abstract.

According to many studies that have been done, it is better to compare

the results of this study with other studies in the results section;

this could be accompanied by adding a table at the end of the results

to make the comparison easier and to determine the superiority of this

study.

Any final polishing of manuscript should be completed prior to

submission and it is strongly recommended to check the text by

professional language editing services to make respective grammar

corrections, especially when English is not the Authors’ native

language.

Reviewer #4: PONE-D-22-29094

Generally, the manuscript should be checked by official English proofread or native.

Abstract

I suggest to add “occupational exposure” as keyword

Add information about number of respondent and human sampling method in abstract.

Introduction

In line 45 – 46, PVC food packaging stretch films, and dental sealants, thermal paper is… change with “PVC food packaging stretch films, and dental sealants, thermal paper are…”

Add citation in statement in line 45 – 46. Why thermal paper is included in the major source of BPA.

Add citation in statement in line 46 – 47

Explain about health effects from BPA exposure in general. Are there any epidemiological studies related BPA exposure from skin to human health problems?

Materials and methods

I suggest to display sampling area map in the manuscript

Why only 30 samples of thermal paper analyzed?

There are 3 constant value for k and AF. can you describe what are for?

Result and discussion

Please change table’s format according to PLOS one format

Conclusion and recommendation

I think your suggestion of sanitizers and lotion application should be reduce is not accepted. Because it will contradict with other regulation related COVID 19 transmission prevention.

6. PLOS authors have the option to publish the peer review history of their article (what does this mean?). If published, this will include your full peer review and any attached files.

Reviewer #1: **Yes: **Waleed K. Abdulsahib

Reviewer #2: No

Reviewer #3: **Yes: **Davoud Balarak

Reviewer #4: No

---

## [Author Response · Author response to Decision Letter 0]

7 Feb 2023

REVIEWER 1:

Comment 1: Abstract is vague, please rewrite to be clear for readers

Response 1: The abstract has been revised for further clarification and to incorporate also Comment 2 raised by Reviewer 3 below

Comment 2: Introduction is too long

Response 2: To find a balance between this comment and comment 1 raised by Reviewer 3 who expressed that Introduction is “too short”, we have revised the Introduction to read more scientifically and shortened few sentences as applicable. Thanks for your valuable comments.

Comment 3: Number of paper is insufficient

Response 3: The study area is Sharjah City (Population 1.275 million (2019)) and number of receipt samples was selected based on literature review and typical number of samples used in similar global studies. For example, 40 sampling location were selected by Khatun et al., 2022 in Dhaka, Bangladesh, 12 samples were collected in duplicates in New Delhi, India (ToxicsLink, 2012), 44 samples were collected from various regions in Belgium by Geens et al., 2012. All such locations have population counts much higher than Sharjah City thus authors adopted similar numbers of thermal paper samples in the current study.

Comment 4: Tables unclear, specially Table 3

Response 4: All Tables have been revised to comply by journal’s format and read clearer

Comment 5: Discussion must be supported by extra references

Response 5: Initially, discussion of results were done in comparison to findings from similar research conducted worldwide where the receipts were specifically collected from supermarkets, restaurants, ATMs and others to be comparable to the sources of thermal receipts in the current study. Additional discussion and references are now included and tabulated (Table 1 and 2) to expand and address the reviewer’s comment.

REVIEWER 2: 

Comment 1: The conclusion should be more specific. It should only contain the important findings. The discussion section can include the comments made in the conclusion.

Response 1: The Conclusion section was revised to include specifically recorded findings. Additional comments and recommendations are moved to the Results and Discussion section as suggested by Reviewer. Thank you for your valuable comments.

Comment 2: It would enrich the study if the study participants' health problems attributable to BPA toxicity were investigated

Response 2: The authors agree that investigation of health impacts attributable to BPA are important yet the proposed aspect is out of scope of the current study as the study aims at investigating BPA levels in thermal receipts circulating in Sharjah City, UAE and conducting related risk assessments. Thus, human monitoring was not included in this study as it is subject to ethical approvals and securing relevant consents which are on the authors’ research agenda. We thank the reviewer on the valuable comment.

Comment 3: There are a few grammatical/typographical mistakes

Response 3: The manuscript was revised with best efforts to eliminate grammatical/typographical mistakes, where applicable.

REVIEWER 3:

The paper presents the results of studies Detection of Bisphenol A in Thermal Paper Receipts and Assessment of Human Exposure: A Case Study from Sharjah, United Arab Emirates. The issue raised in the paper is important and the manuscript falls within the scope of PLOS one. The paper would be of interest to the readers of the journal; however, I recommend this paper undergo major revisions. The results are quite interesting but still the flaws are needed to be addressed.

Comment 1: Introduction is too short and simple and written in very simple and unscientific way

Response 1: This comment, although in contradiction with Reviewer 1 Comment 2, was addressed by revising the Introduction to read more scientifically and shortened few sentences as applicable

Comment 2: I think that at the end of the abstract should be inserted a brief conclusion to help readers in getting a better understanding of what is the significance of this study. Quantitative information should be provided in the abstract.

Response 2: Quantitative information such as BPA levels, exceedances, and estimated daily intakes were made sure to be included in the abstract, and the “Abstract” section was concluded by a phrase highlighting the significance of the current study

Comment 3: According to many studies that have been done, it is better to compare the results of this study with other studies in the results section; this could be accompanied by adding a table at the end of the results to make the comparison easier and to determine the superiority of this study.

Response 3: Initially, discussion of results were done in comparison to findings from similar research conducted worldwide where the receipts were specifically collected from supermarkets, restaurants, ATMs and others to be comparable to the sources of thermal receipts in the current study. Additional discussion and references are now included and tabulated (Table 1 and 2) to expand and address the reviewer’s comment. Thank you for your valued comment.

Comment 4: Any final polishing of manuscript should be completed prior to submission and it is strongly recommended to check the text by professional language editing services to make respective grammar corrections, especially when English is not the Authors' native language.

Response 4: The manuscript was revised with best efforts to eliminate grammatical/typographical mistakes, where applicable.

REVIEWER 4: 

Comment 1: I suggest to add "occupational exposure" as keyword; add information about number of respondent and human sampling method in abstract.

Response 1: “Occupational exposure” is added as a keyword. Human sampling was out of the scope of the current study which primarily aims at investigating BPA levels in thermal receipts circulating in Sharjah City, UAE and assessing related risks rather than investigating human subjects and conducting surveys. Thank you for your valuable comment

Comment 2: In line 45 - 46, PVC food packaging stretch films, and dental sealants, thermal paper is... change with "PVC food packaging stretch films, and dental sealants, thermal paper are..."

Response 2: Sentence (now line 54) corrected as suggested, thank you.

Comment 3: Add citation in statement in line 45 - 46. Why thermal paper is included in the major source of BPA.

Response 3: Citation included (now in line 54) as suggested

Comment 4: Add citation in statement in line 46 – 47. Explain about health effects from BPA exposure in general. Are there any epidemiological studies related BPA exposure from skin to human health problems?

Response 4: Additional citations are included as suggested. Health effects from BPA exposure are also discussed in now line 51. Thank you for your valuable comment

Comment 5: I suggest to display sampling area map in the manuscript. Why only 30 samples of thermal paper analyzed?

Response 5: The study area is Sharjah City (Population 1.275 million (2019)) and number of receipt samples was selected randomly based on literature review and typical number of samples used in similar global studies. For example, 40 sampling location were selected by Khatun et al., 2022 in Dhaka, Bengladesh, 12 samples were collected in duplicates in New Delhi, India (ToxicsLink, 2012), 44 samples were collected from various regions in Belgium by Geens et al., 2012. All such locations have population counts much higher than Sharjah City so authors adopted similar numbers of thermal paper samples in the current study. It is noteworthy to mention that higher number of samples would have been desired yet incur higher laboratory efforts and costs. Thank you for the valuable comment

Comment 6: There are 3 constant value for k and AF. can you describe what are for?

Response 6: Constant values for k and AF were selected from the extensive literature review conducted throughout the study. The authors made efforts to investigate all possible reported values for k, paper-to-skin transfer coefficients of BPA and all reported absorption fractions of BPA by skin (AF) as reported by Bernier and Vanderberg (2017) for a comprehensive risk assessment covering various scenarios. 

Comment 7: Please change table's format according to PLOS one format

Response 7: Noted with thanks, all Tables have been revised to comply by journal’s format. 

Comment 8: I think your suggestion of sanitizers and lotion application should be reduced is not accepted. Because it will contradict with other regulation related COVID 19 transmission prevention.

Response 8: Noted with thanks, the revised recommendation calls occupationally exposed persons to wear gloves, especially with sanitizers and lotion application during COVID-19 pandemic, to reduce exposure to BPA from thermal papers.

---

## [Decision Letter · Decision Letter 1]

28 Feb 2023

PONE-D-22-29094R1Detection of Bisphenol A in Thermal Paper Receipts and Assessment of Human Exposure: A Case Study from Sharjah, United Arab EmiratesPLOS ONE

Dear Dr. Semerjian,

Thank you for submitting your manuscript to PLOS ONE. After careful consideration, we feel that it has merit but does not fully meet PLOS ONE’s publication criteria as it currently stands. Therefore, we invite you to submit a revised version of the manuscript that addresses the points raised during the review process.

We look forward to receiving your revised manuscript.

Kind regards,

Yasmina Abd‐Elhakim

Academic Editor

PLOS ONE

Journal Requirements:

**Reviewers' comments:**

**Reviewer #1:** (No Response)

**Reviewer #2: **The reviewers comments are well addressed.

However, it is preferable to keep the conclusion and recommendation separate.

**Reviewer #3: **Thanks for your correct revision, and the quality of the paper has been improved greatly so that it is adequate to publish in journal.

**Reviewer #4: **please to explain about 3 numbers of k (paper-to-skin transfer coefficient) and AF (absorption factor) that you display on table 3 - 7 ? why there are 3 numbers? what are they for? Is the number differentiated based on the age of the respondent or otherwise? you can add the information on the text or in table.

TDI is Tolerable Daily Intake not Tolerable day intake (please review the text).

---

## [Author Response · Author response to Decision Letter 1]

8 Mar 2023

Reviewers' comments:

Reviewer #1:

No response/comments received.

Reviewer #2: 

Comment: The reviewers’ comments are well addressed. However, it is preferable to keep the conclusion and recommendation separate.

Response: Thank you, as suggested the “Conclusions” have been separated from the “Recommendations” in the revised version

Reviewer #3:

Comment: Thanks for your correct revision, and the quality of the paper has been improved greatly so that it is adequate to publish in journal.

Response: Thank you for your positive response

Reviewer #4:

Comment: please to explain about 3 numbers of k (paper-to-skin transfer coefficient) and AF (absorption factor) that you display on table 3 - 7 ? why there are 3 numbers? what are they for? Is the number differentiated based on the age of the respondent or otherwise? you can add the information on the text or in table.

Response: Thank you. Further explanations for both k and AF, namely as below, were added in-text to the “Materials and Methods: BPA exposure assessment” section (p. 8, Lines 167-186, and 172-175 in Revised Manuscript with Track Changes).

For k values: Reported k values varied as conducted extractability experiments exhibited varying coefficients based on skin conditions being dry, wet, or very greasy.

For AF values: A wide range of absorption factors have been reported in previous studies as they are calculated from different experimental systems: 2.3–8.6% from skin explants, and 27% from live hands after application of BPA

Comment: TDI is Tolerable Daily Intake not Tolerable day intake (please review the text).

Response: The text has been reviewed, the phrase now reading Tolerable Daily Intake (p. 14, Line 253 in Revised Manuscript with Track Changes)

---

## [Editor Report · Decision Letter 2]

14 Mar 2023

Detection of Bisphenol A in Thermal Paper Receipts and Assessment of Human Exposure: A Case Study from Sharjah, United Arab Emirates

PONE-D-22-29094R2

Dear Dr. Semerjian,

We’re pleased to inform you that your manuscript has been judged scientifically suitable for publication and will be formally accepted for publication once it meets all outstanding technical requirements.

Kind regards,

Yasmina Abd‐Elhakim

Academic Editor

PLOS ONE
---

## [Editor Report · Acceptance letter]

20 Mar 2023

PONE-D-22-29094R2 

Detection of bisphenol A in thermal paper receipts and assessment of human exposure: A case study from Sharjah, United Arab Emirates 

Dear Dr. Semerjian:

I'm pleased to inform you that your manuscript has been deemed suitable for publication in PLOS ONE. Congratulations! Your manuscript is now with our production department. 

Kind regards, 

on behalf of

Dr. Yasmina Abd‐Elhakim 

Academic Editor

PLOS ONE